# Circulating microRNAs as Biomarkers of Hepatic Fibrosis in Schistosomiasis Japonica Patients in the Philippines

**DOI:** 10.3390/diagnostics12081902

**Published:** 2022-08-05

**Authors:** Ian Kim B. Tabios, Marcello Otake Sato, Ourlad Alzeus Gaddi Tantengco, Raffy Jay C. Fornillos, Masashi Kirinoki, Megumi Sato, Raniv D. Rojo, Ian Kendrich C. Fontanilla, Yuichi Chigusa, Paul Mark B. Medina, Mihoko Kikuchi, Lydia R. Leonardo

**Affiliations:** 1Institute of Biology, College of Science, University of the Philippines Diliman, Quezon City 1101, Philippines; 2College of Medicine, University of the Philippines Manila, Ermita, Manilla 1000, Philippines; 3Laboratory of Tropical Medicine and Parasitology, Dokkyo Medical University, 880 Kitakobayashi, Mibu-machi, Shimotsuga-gun, Tochigi 321-0293, Japan; 4Graduate School of Health Sciences, Niigata University, 2-746 Asahimachi-dori, Chuo-ku, Niigata City 951-8518, Japan; 5Center for International Cooperation, Dokkyo Medical University, 880 Kitakobayashi, Mibu-machi, Shimotsuga-gun, Tochigi 321-0293, Japan; 6Department of Biochemistry and Molecular Biology, College of Medicine, University of the Philippines Manila, Ermita, Manilla 1000, Philippines; 7Institute of Tropical Medicine, Nagasaki University, 1-12-4 Sakamoto, Nagasaki 852-8523, Japan; 8University of the East Office of Research Coordination, Manila 1008, Philippines; 9College of Arts and Sciences, University of the Philippines Manila, Ermita, Manilla 1000, Philippines; 10University of the East Ramon Magsaysay Graduate School, Quezon City 1100, Philippines

**Keywords:** biomarker, miRNA, prognosis, schistosome, *Schistosoma japonica*

## Abstract

Host-derived microRNAs (miRNAs) play important regulatory roles in schistosomiasis-induced hepatic fibrosis. This study analyzed selected serum miRNAs among Filipino schistosomiasis japonica patients with ultrasound (US)-detectable hepatic fibrosis. A prospective cohort study design with convenience sampling was employed from 2017 to 2019. The study sites were eight endemic barangays in Leyte, Philippines. Eligible chronic schistosomiasis patients with varying severities of hepatic fibrosis were enrolled in the cohort and serially examined at 6, 12, and 24 months from baseline. Baseline serum miR-146a-5p, let-7a-5p, miR-150-5p, miR-122-5p, miR-93-5p, and miR200b-3p were measured using RT-qPCR. A total of 136 chronic schistosomiasis patients were included in this prospective cohort study. Approximately, 42.6% had no fibrosis, 22.8% had mild fibrosis, and 34.6% had severe fibrosis at baseline The serum levels of the antifibrotic miR-146a (*p* < 0.0001), miR-150 (*p* = 0.0058), and let-7a (*p* < 0.0001) were significantly lower in patients with hepatic fibrosis while the profibrotic miR-93 (*p* = 0.0024) was elevated. miR-146a-5p (AUC = 0.90, 95% CI [0.84, 0.96], *p* < 0.0001) has the most promising potential to differentiate patients with (*n* = 78) versus without (*n* = 58) hepatic fibrosis. The baseline level of serum miR-146-5p was significantly different in patients with progressive fibrosis (*n* = 17) compared to those who never developed fibrosis (*n* = 30, *p* < 0.01) or those who had fibrosis reversal (*n* = 20, *p* < 0.01) after 24 months. These findings demonstrate the potential utility of serum miRNAs, particularly of miR-146a, as a supplementary tool for assessing hepatic fibrosis in chronic schistosomiasis japonica patients.

## 1. Introduction

Schistosomiasis has always been one of the most prevalent helminthic diseases causing huge socioeconomic and public health impacts [1]. In the Philippines, schistosomiasis remains endemic in 81 provinces [2]. Around 12 million people are at risk of schistosomiasis japonica with 2.5 million directly exposed to snail-infested bodies of water [3]. This is associated with irrigation networks that harbor intermediate snail vectors of the parasite [4,5]. Despite efforts to control its spread, schistosomiasis remains to be classified as a neglected tropical disease concentrated in underserved communities in developing nations [6].

Although the prevalence and infection intensity of schistosomiasis have significantly declined because of regular mass drug administration (MDA) with praziquantel (PZQ), clinical morbidities due to chronic schistosomiasis still afflict a significant number of individuals. Hepatosplenic schistosomiasis, the primary cause of morbidity and mortality in chronically infected individuals, is present in up to 50% of residents in highly endemic areas of the Philippines [7]. It typically manifests with hepatic fibrosis without bridging, nodular formation, or significant hepatocellular destruction [8]. Mild cases are primarily asymptomatic and have normal liver biochemical tests. Signs and symptoms reported in advance cases are secondary to the development of portal hypertension and can include hematemesis, melena, ascites, pallor, ankle edema, and collateral periumbilical varices. In the absence of other inciting factors, isolated hepatosplenic schistosomiasis rarely leads to end-stage liver disease [9].

Ultrasound (US) is the most widely utilized method for detecting hepatosplenic pathology in endemic areas. It is a reliable tool for evaluating the severity of schistosomiasis-induced hepatosplenic abnormalities and for monitoring treatment response [10]. A distinct feature seen in *S. japonicum*-infected patients is the presence of mosaic or network echogenic pattern. This US image is not observed in patients affected by other schistosome species [11]. Grading systems have been developed to correlate disease burden with US findings [12]. Despite inter-observer variability, the usefulness of US in the evaluation of chronic schistosomiasis has been demonstrated in various studies for the past 30 years [13].

Studies on schistosomiasis have also utilized circulating miRNAs in body fluids to elucidate disease pathogenesis to circumvent difficulties in obtaining liver samples. Measurement of six miRNAs (miR-146b, miR-122, miR-223, miR-199a, and miR-34a) in sera from mice, rabbits, buffalos, and humans showed that serum miR-223 was elevated in infected patients as compared to healthy controls. Moreover, miR-233 returned to normal levels in mice sera after PZQ treatment. This highlights the potential use of host-derived miRNAs as novel biomarkers for detecting active infection and the assessment of PZQ response [14]. Cai et al. (2015) identified circulating serum miR-122, miR-21, and miR-34a as possible biomarkers for the progression of hepatic fibrosis in schistosomiasis-infected BALB/c mice [15]. In a cohort of Filipino patients, serum miR-150, let-7a, let-7d, miR-146a, and serum exosomal miR-146a differentiated patients with mild versus severe fibrosis [16,17]. These studies demonstrated the potential utility of circulating miRNAs as biomarkers for diagnosis and prognostication of schistosomiasis-induced liver pathology.

Many individuals residing in endemic areas still present with hepatosplenic schistosomiasis and are at risk of developing hepatic fibrosis. This study determined the persistence, reversal, and progression of US-detectable hepatic fibrosis among chronic schistosomiasis patients at 6, 12, and 24 months after the initial examination. Moreover, this study determined the role of selected serum miRNAs among Filipino schistosomiasis japonica patients with different degrees of ultrasound (US)-detectable hepatic fibrosis.

## 2. Materials and Methods

### 2.1. Ethical Considerations

This study was approved by the University of the Philippines Manila Research Ethics Board UPMREB Code 2017-369-01 on 27 November 2017. Written informed consent was obtained from all participants at the beginning of the study.

### 2.2. Study Area and Population

The study sites comprised eight schistosomiasis endemic barangays in the municipalities of Julita, Alang-alang, Palo, and Sta Fe in the province of Leyte. These areas are low land farmlands and non-endemic for malaria based on the Department of Philippines (DOH) database and communication with their respective municipal health officers (MHO). In 2015, the prevalence of schistosomiasis in one of these municipalities, Alang-alang, was 6.01%. [18]. There have been schistosomiasis control programs with annual mass drug administration in these municipalities since 2008 [7]. Using a pre-tested semi-structured schistosomiasis questionnaire, face-to-face interviews were performed by trained barangay health workers (Appendix A). Sample collection was performed before MDA in the localities to capture more stool-positive individuals. The manuscript complies with the STROBE reporting guidelines for observational studies.

### 2.3. Study Design and Sampling Technique

A community-based prospective cohort study design and a convenience sampling technique were employed. Residents who were conveniently available during the fieldwork were invited to the study. An initial cross-sectional study was performed to identify chronic schistosomiasis patients who were invited to the cohort study.

### 2.4. Sample Size

The sample size was computed based on sample size calculation in studies focusing on expression profile analysis using Poisson distribution. To detect differentially expressed genes with at least a fold change of 1.5 in single-gene testing (α = 0.05, 1−β = 0.8) between cases and controls, 126 or more schistosomiasis japonica patients were targeted to be included in the prospective longitudinal study [19]. The updated prevalence (2016) using sentinel stool surveys in the highly endemic barangays of Alangalang ranged from 20.1 to 25.6% based on personal communication with the Schistosomiasis Control and Elimination Program (SCEP) Regional Coordinator for Eastern Visayas Region. Using an estimated prevalence of 20% and precision of 0.05, at least 715 individuals in all municipalities should be screened during the initial survey to identify at least 126 cases (http://sampsize.sourceforge.net/iface/index.html (accessed on 15 March 2017)).

### 2.5. Eligibility Criteria

The inclusion criteria for the prospective cohort study were the following: (1) positive on both Kato–Katz (K-K) and soluble egg antigen (SEA) IgG ELISA at the beginning of the study; (2) completed PZQ (60 mg/kg) treatment within two weeks after diagnosis; (3) age between 18 and 49 years old; (4) resides primarily in the study area for more than ten years based on the interview; (5) presence of significant exposure to potentially contaminated water based on the interview; (6) provided enough blood (at least 10 mL) for molecular and biochemical analyses at baseline; and (7) had ultrasound examination result at baseline. The exclusion criteria were the following: (1) presence of cirrhosis and/or ascites based on clinical and US findings at baseline; (2) history or positive serology for hepatitis virus types B and C; (3) severely underweight (BMI < 16.00) or extremely obese (BMI > 40.0); (4) diagnosed with other liver disease; (5) known case of cancer; (6) pregnant and/or lactating; (7) evidence of US-detectable hepatosplenic pathology not commonly seen in schistosomiasis; and (8) ≥60 g per day of ethanol consumption for ≥10 years based on interview. Withdrawal criteria included (1) withdrawing informed consent, (2) changing of residence, (3) developing other liver diseases, or (4) becoming pregnant. 

### 2.6. Prospective Study

Based on the eligibility criteria for the prospective study, 159 patients were enrolled in the cohort, of which 136 had complete data for the final analysis (Figure 1). In the prospective study, follow-up surveys of the enrolled participants were performed 6, 12, and 24 months after the initial survey. A comparison was made with the results at the beginning of the study and 6-, 12-, and 24-month follow-up surveys (Appendix A).

### 2.7. Ultrasound and Clinical Evaluation

The US procedure was performed using portable Vscan Pocket Ultrasound (GE Healthcare, Chicago, IL, USA) based on the protocol of Ohmae et al. (1992) with minor changes in accordance with the WHO recommendations [11,12]. Two trained personnel conducted the US examinations and interpretation at all time points. Liver and spleen organometries were based on reference measurements among a healthy Chinese population due to the lack of data from healthy Filipinos (Appendix A). Based on the US-detected echogenic pattern and the thickness of the portal vein wall, liver pictures were divided into three categories: (1) type 0 or within normal limits (WNL) exhibiting no echogenic patterns and absence of portal vein wall thickening; (2) types 1 and 2 or mild fibrosis exhibiting linear or tubular echogenic bands with mild, moderate, or severe echogenic thickening (6 mm) of the portal vein wall; and (3) type 3 network pattern exhibiting septal formation into regular geometric blocks, with more than three blocks surrounded by high echogenic bands (Appendix A) [7]. 

### 2.8. SEA IgG ELISA

SEA Ig ELISA was performed with some minor modifications [20,21]. Rabbit anti-human IgG-peroxidase antibody was used as the secondary antibody, while 3,3′,5,5′-tetramethylbenzidine (SeraCare, Milford, MA, USA) served as the substrate. SEA (1 g/well) was coated on polystyrene 96-well ELISA plates (Greiner Bio-One, Frickenhausen, Germany). Antigen was diluted with 0.05 M carbonate bicarbonate buffer (pH 9.6). Serum samples were dispensed onto the antigen-coated wells after blocking with % bovine serum albumin (BSA) in phosphate-buffered saline with 0.05% Tween 20 (T-PBS) (T-PBS-1% BSA). A 200-fold and 10,000-fold dilution of human sera (0.1 mL) and secondary antibody (0.1 mL) were used, respectively. Optical density (OD) at 450 nm was measured using a microplate reader. Each ELISA reaction utilized positive (8 weeks post-infected rabbit serum) and negative (diluting buffer) controls. The experiments were performed in triplicate. The cut-off value was the mean absorbance of sera from healthy controls plus three standard deviations. Positive samples were those whose mean absorbance was higher than the cut-off value.

### 2.9. Target miRNA Identification

Target miRNAs were identified based on a literature review using the PubMed search engine and miRbase (Appendix A). Published articles regarding liver fibrosis and miRNA were gathered. The following filters were manually applied to select targets to be included in this study: (1) miRNAs from any biological sample dysregulated in liver fibrosis caused by any hepatic pathology; (2) miRNAs from any biological sample dysregulated in schistosomiasis-related liver fibrosis in animal models; (3) miRNAs found in detectable amounts in the sera. A representative miRNA from different molecular pathways involved in liver fibrosis was purposively selected.

### 2.10. RNA Extraction

Total RNA was extracted from 100 µL serum using the miRNeasy mini kit (Qiagen, Hilden, Germany) according to the manufacturer’s protocol. RNA Spike-In kit (Exiqon, Vedbaek, Denmark) containing *Caenorhabditis elegans* cel-miR-39-3p miRNA mimic was added to each sample. The final RNA product was eluted into 30 μL nuclease-free water and stored at −80 °C before further analysis. RNA quantity and quality were assessed using a NanoDrop 1000 spectrophotometer (Thermo Fisher Scientific, Waltham, MA, USA). Only RNA extracts with 260 nm/280 nm absorbance ratio > 1.8 were utilized in the downstream processes. 

### 2.11. Reverse Transcription PCR (RT PCR)

First-strand cDNA synthesis was performed using Universal cDNA synthesis kit II (Exiqon, Vedbaek, Denmark) following the manufacturer’s protocol in a 10 μL RT reaction: 2 μL 5× reaction buffer, 1 μL enzyme mix, 0.5 μL synthetic RNA spike in UniSp6 (0.075 fmol/μL or 5 × 10^7^ copies/μL), 4.5 μL nuclease-free water, and 2 μL template total RNA. Reverse transcription (RT) reactions were conducted using a thermal cycler under the following conditions: 42 °C for 60 min, 95 °C for 5 min. RT-PCR products were immediately stored undiluted at −20 °C prior to the further qPCR testing.

### 2.12. Initial miRNA Profiling

Following the manufacturer’s instructions, initial miRNA profiling was performed with miScript miRNA PCR array human miRNome (96-well plate) (Qiagen, Hilden, Germany). Four (4) sets of pooled sera were used: (1) chronic schistosomiasis patients without hepatic fibrosis (8 sera); (2) chronic schistosomiasis patients with mild hepatic fibrosis (8 sera); (3) chronic schistosomiasis patients with severe hepatic fibrosis (8 sera); and (4) non-infected participants without hepatic fibrosis (8 sera). 

### 2.13. Validation of miRNA Expression Using Quantitative PCR (qPCR)

Potential target miRNAs based on literature review and initial miRNA profiling were used in this study. LNATM (Exiqon, Vedbaek, Denmark) miRNA-specific primers were utilized. miRNA quantification was performed using a miRCURY LNA TM Universal RT microRNA PCR (Exiqon, Vedbaek, Denmark) kit following manufacturer’s protocol in 10 μL PCR reaction containing 5 μL PCR master mix, 1 μL PCR primer mix (final concentration 0.2 μM), and 4 μL diluted cDNA template (final concentration 1:140). ROX (Thermo Fisher Scientific, Waltham, MA, USA), a passive reference dye, was added in the diluted cDNA template at a 1: 20,000 final concentration. Amplification was performed on an ABI PRISM ^®^ 7700 (Thermo Fisher Scientific, Waltham, MA, USA) with the cycling conditions: polymerase activation and template denaturation at 95 °C for 10 min, followed by 40 cycles of 95 °C for 10 sec, and 60 °C for 60 s at ramp rate 1.6 °C/s. 

The mean of the endogenous SNORD95 and the spiked-in UniSp6 was used for appropriate normalization. Using reference miRNAs in the normalization of qPCR data highlights actual biological differences among the test samples while reducing the effects of variability incurred during experimental procedures [22]. Relative quantification (*RQ*) and expression levels were calculated using the following formulas [23]: (1)$$Expression=2^{\;\Delta C_{t}}$$
(2)$$RQ=2^{\;\Delta \Delta C_{t}}$$
where *Ct*, or threshold cycle, is the PCR cycle number at which the fluorescence level meets the specified threshold value; Δ*Ct* = *Ct* target miRNA − *Ct* reference miRNA; and ΔΔ*Ct* = Δ*Ct* test sample − Δ*Ct* calibrator sample. 

Average *Ct* values of SNORD95 and UniSp6 were utilized as the *Ct* reference. Sera from either non-infected or infected individuals without hepatic fibrosis were used as calibrator samples. Based on the manufacturer’s recommendation, *Ct* values less than 32 were considered true amplification, while *Ct* values more than 32 were considered as late amplifications and excluded from the analysis. Triplicates were performed for each sample.

### 2.14. Target Predictions

miRWalk 2.0 (zmf.umm.uni-heidelberg.de/apps/zmf/mirwalk2/path-self.html (accessed on 10 December 2020)) and TargetScan 7.0 (www.targetscan.org (accessed on 12 December 2020)) were used to predict possible targets of the selected miRNAs, which might play a role in hepatic fibrosis. The functions and putative disease relations of the predicted target genes were investigated using UniProt (www.uniprot.org/ (accessed on 12 December 2020)) and NCBI (www.ncbi.nlm.nih.gov/gene (accessed on 12 December 2020)).

### 2.15. Data Analyses Plan

The changes in the severity of hepatic fibrosis at the beginning of the study and at 6, 12, and 24 months post-PZQ were compared among the groups. The following criteria were used in comparing hepatic fibrosis at baseline and at follow-up: (1) progressive fibrosis (PF), defined as a higher grade of hepatic fibrosis based on US findings at the follow-up period compared to baseline; (2) reversal of fibrosis (RF), defined as a lower grade of hepatic fibrosis based on US findings at the follow-up period compared to baseline; (3) stable US findings (SF), defined as the same US findings during baseline and the subsequent follow-up periods. Other US abnormalities were also compared [7]. 

### 2.16. Statistical Analyses

Comparisons between groups for quantitative data were performed using the chi-square test. One-way ANOVA was used for within-group comparisons for continuous variables. ROC curves were performed to describe the diagnostic performance of the selected miRNAs differentially expressed among groups. The area under the curve (AUC) was interpreted as follows: AUC of 0.5 suggests no discrimination; AUC of 0.7 to 0.8 is acceptable; AUC of 0.8 to 0.9 is excellent; and AUC > 0.9 is outstanding [24]. Double-entered data stored in EpiData 3.1 were analyzed using GraphPad Prism version 9 (GraphPad Software, San Diego, CA, USA) for Mac. A *p*-value of <0.05 was considered statistically significant.

## 3. Results

### 3.1. Characteristics of Patients

The prospective study was conducted from November 2017 to November 2019 with three follow-up studies performed 6 (June 2018), 12 (November 2018), and 24 (November 2019) months after PZQ treatment, which was given within 1–2 weeks after the initial data collection. Figure 1 shows the participant recruitment and follow-up diagram. There were 136 patients with complete data available for analysis. Among the 136 participants with complete data for analysis, 58, 31, and 47 patients were classified at baseline as Type 0 or no fibrosis, Types 1–2 or mild fibrosis, and Type 3 or severe fibrosis, respectively. 

Table 1 presents the demographic and basic clinical profiles at baseline of the 136 patients included in the prospective study analysis. Only egg per gram (EPG) was significantly different among groups (Brown–Forsythe ANOVA test, F* = 11.50, *p* < 0.001), with Type 3 patients having higher EPG compared to Type 0 (Dunnett’s T3 multiple comparisons test, *p* = 0.0005) and Types 1–2 (Dunnett’s T3 multiple comparisons test, *p* = 0.0044) patients. The difference in EPG between Type 0 and Types 1–2 was not significant (Dunnett’s T3 multiple comparisons test, *p* = 0.8203).

### 3.2. Evaluation of Hepatic Fibrosis Outcomes by Ultrasonography

All 136 patients had complete data on focused US, stool examination, and serological testing at baseline and 6, 12, and 24 months post-PZQ (Table 2). There were no stool-positive participants at the 6- and 12-month follow-up. After 24 months, re-infection based on stool exams in 34 patients (25.0%) was detected, with low and moderate infection in 32 and 2 patients, respectively. Despite negative stool examination, most patients remained seropositive on SEA IgG ELISA. 

Among the detected US abnormalities, only the proportion with splenomegaly was significantly decreased at 6 months post-PZQ (Fisher’s exact test, two-sided, *p* = 0.0064). The proportion of patients with hepatomegaly was significantly lower than baseline starting at 12 months post-PZQ (Fisher’s exact test, two-sided, *p* = 0.0256). Even after 24 months, the proportion of patients with periportal fibrosis did not significantly decrease (Fisher’s exact test, two-sided, *p* = 0.0617).

Appendix A and Figure 2 show the dynamics of hepatic fibrosis severity at different time points. There were 58 patients without hepatic fibrosis at baseline, of which 46 (79.3%) remained without hepatic fibrosis after 24 months while the other 12 (20.7%) progressed to mild fibrosis. At baseline, 47 patients with severe fibrosis had stable fibrosis with the same type 3 US findings even at 24 months follow-up. The 31 patients with mild fibrosis at baseline had more diverse outcomes after 24 months, with 6 (19.4%), 5 (16.1%), and 20 (64.5%) having stable fibrosis (remained with mild), progressive fibrosis (mild to severe fibrosis), and reversal of fibrosis (mild to no fibrosis), respectively. Most patients (72.8%), regardless of baseline results, had stable US findings, while 12.5% had progressive fibrosis and 14.7% had a reversal of fibrosis after 24 months.

### 3.3. Selection of miRNAs

Using the human miRNOme miRNA PCR array, 102 out of the 1008 (10.1%) most abundantly expressed and best-characterized miRNAs were detected in the serum samples. Among the 102 detectable miRNAs, 19 (18.6%) had statistically significant fold change of either ≥2 or ≤0.5 between chronic schistosomiasis patients with and without hepatic fibrosis. Eleven miRNAs (miR-122-5p, miR-200b-3p, miR-146a-5p, miR-150-5p, miR-93-5p, let-7a-5p, miR-151a-5p, miR-21-5p, miR-27a-3p, miR-16-5p, let-7i-5p) were selected for further analysis based on their statistically significant fold regulation observed in the initial profiling and their putative involvement in fibrogenesis and fibrosis reversal (Appendix A). Of the eleven miRNAs, only six (6) were consistently detected in the individual sera using RT-qPCR. These were miR-122-5p, miR-200b-3p, miR-146a-5p, miR-150-5p, miR-93-5p, and let-7a-5p. Subsequent analyses only involved these six target miRNAs. Average Ct values of the endogenous SNORD95 and spiked-in UniSp6 were used to normalize the data.

### 3.4. RT-qPCR Validation of miRNAs

Baseline serum levels of the 6 miRNAs were measured in the following groups: (1) Sj−/T0 (*n* = 37)—K-K and SEA ELISA negative participants without hepatic fibrosis; (2) Sj+/T0 (*n* = 58)—K-K and SEA ELISA positive participants without hepatic fibrosis; (3) Sj+/T1−2 (*n* = 31)—K-K and SEA ELISA positive participants with mild hepatic fibrosis; and (4) Sj+/T3 (*n* = 47)—K-K and SEA ELISA positive participants with severe hepatic fibrosis. To check if these miRNAs can be used to detect active infection independent of US status, expression levels between Sj+/T0 and S−/T0 were compared (Appendix A). All six miRNAs were not significantly different between those with and without active infection (miR-146a-5p, *p* = 0.6473; let-7a-5p, *p* = 0.0651; miR-150-5p, *p* = 0.0860; miR-93-5p, *p* = 0.8035; miR-122-5p, *p* = 8.359; miR-200b-3p, *p* = 0.2174).

### 3.5. miRNAs as Biomarkers of Hepatic Fibrosis

Appendix A presents a comparison of miRNAs between chronic schistosomiasis patients with (Sj+/T1–3) and without (Sj+/T0) hepatic fibrosis. Serum levels of the antifibrotic miR-146a-5p (*p* < 0.0001), let-7a-5p (*p* < 0.0001), and miR-150-5p (*p* = 0.0074) were significantly lower in patients with hepatic fibrosis, while the profibrotic miR-93-5p (*p* = 0.0003) was significantly elevated. The antifibrotic miR-122-5p (*p* = 0.1777) and the profibrotic miR-200b-3p (*p* = 0.8867) were not significantly different between the study groups. To check if the four differentially expressed miRNAs can be used for stage-specific diagnosis, chronic schistosomiasis patients with mild (Sj+/T1–2) and severe (Sj+/T3) hepatic fibrosis were compared. Serum miR-146a-5p (*p* = 0.7406), let-7a-5p (*p* = 0.3927), miR-150-5p (*p* = 0.0220), and miR-93-5p (*p* = 0.9969) were not significantly different between those with mild versus severe fibrosis (Appendix A).

Based on ROC curve analysis of the differentially expressed circulating serum miRNAs (Figure 3 and Table 3), miR-146a-5p (AUC = 0.9003, 95% CI [0.8436, 0.9570], *p*-value < 0.0001) had the most potential to discriminate chronic schistosomiasis patients with versus without hepatic fibrosis, followed by let-7a-5p (AUC = 0.7250, 95% CI [0.6395, 0.8109], *p*-value < 0001), miR-93-5p (AUC = 0.6687, 95% CI [0.5738, 0.7635], *p*-value = 0.0008), and miR-150-5p (AUC = 0.6338, 95% CI [0.5402, 0.7274], *p*-value = 0.0077). Table 3 indicates the expression level cut-offs of the differentially expressed miRNAs and their corresponding diagnostic performance based on ROC curve optimal cut-off analysis. At an expression level cut-off of <−8.16, serum miR-146a-5p can discriminate chronic schistosomiasis patients with versus without hepatic fibrosis at sensitivity of 60.3% (95% CI [47.5, 71.9]), specificity of 98.7% (95% CI [93.1, 99.9]), and LR of 47.1.

### 3.6. miRNAs as Biomarkers of Hepatic Fibrosis Outcomes

To check the association between baseline serum levels of the six target miRNAs and US-based hepatic fibrosis outcomes, the following groups were compared: (1) SF (*n* = 30)—patients without a change in US findings; (2) RF (*n* = 20)—patients with reversal of hepatic fibrosis; and (3) PF (*n* = 17)—patients with progressive hepatic fibrosis (Appendix A). The four differentially expressed miRNAs were further analyzed. Moreover, those with severe fibrosis at baseline were not included in the analysis since they all remained with severe fibrosis after the follow-up period. Of the four miRNAs, only baseline serum miR-146-5p was significantly different between patients who had worse US outcomes of progressive fibrosis compared to those with better US outcomes of stable US findings (*p* < 0.01) and reversal of fibrosis (*p* < 0.01) after 24 months. Based on ROC curve analysis (Figure 4), baseline serum miR-146a-5p can differentiate those with poorer US outcomes from those with better US outcomes at AUC of 0.7941 (95% CI [0.6384, 0.9498], *p* = 0.0023) for PF versus SF and 0.7754 (95% CI [0.6138, 0.9352], *p* = 0.0019) for PF versus RF.

## 4. Discussion

Despite the improvements in morbidity and mortality due to schistosomiasis, many individuals residing in endemic areas, both K-K positives and negatives, have subclinical US-detectable hepatosplenic changes that predispose them to develop other liver diseases [10]. In this study, 23.7% of patients examined at baseline had hepatic parenchymal fibrosis, all of whom do not have signs and symptoms consistent with advanced hepatosplenic schistosomiasis (HSS). This is relatively lower than the recent reports showing almost half of patients having fibrosis. This difference in the proportion of afflicted patients is expected. Moreover, the current study was conducted in barangays with higher MDA compliance than in areas in Northern Samar, where past research examining chronic schistosomiasis patients using US was carried out [10,25,26]. 

### 4.1. Serological Markers of Liver Status

In this study, serological markers of hepatic function supported the observation in previous research that most HSS patients with no signs of clinical decompensation have minimal biochemical evidence of hepatocellular dysfunction [11]. The serum liver transaminases are expected to be not significantly different between infected patients and healthy controls and among infected patients with varying degrees of hepatic fibrosis. Hepatocellular integrity is preserved in most chronic schistosomiasis patients without concomitant liver disease despite US-detectable hepatic fibrosis. The result of this study is consistent with previous studies that levels of serum ALT and AST are not a good candidate as a biochemical marker for the diagnosis and grading of US-detectable subclinical hepatic fibrosis among schistosomiasis patients in the absence of other diseases affecting the liver such as viral hepatitis, alcoholic liver disease, and non-alcoholic fatty liver disease [7,11,27,28].

### 4.2. Hepatic Fibrosis Outcomes

Aside from baseline examination, this study investigated the post-PZQ treatment history of US-detectable hepatosplenic pathology within 24 months of follow-up among the 136 chronic schistosomiasis patients without evidence of other hepatic diseases. All patients participated in the yearly MDA (40 mg/kg) on top of the PZQ treatment (60 mg/kg) baseline [29]. Consistent with previous findings in longitudinal studies involving post-PZQ treatment, most study participants with mild fibrosis at baseline had regression of fibrosis grade, with 64.5% having within normal US liver findings (type 0) after 24 months from the initial examination. Despite the similar exposure to PZQ treatment during the study duration, patients with severe fibrosis had stable fibrosis with no qualitative US evidence of hepatic fibrosis reversal, even at the final follow-up at 24 months. Ohmae et al. (1992) observed the same trend wherein patients with severe fibrosis had no US-detectable improvement in fibrosis grade after 17 months from PZQ treatment as compared to those with mild fibrosis, of which fibrosis reversal beginning at 6 months post-PZQ was observed in most individuals [11].

Similarly, Olveda et al. (2017) did a longitudinal study on highly endemic barangays in Northern Samar using a different US-based grading scheme for schistosomiasis-related hepatic fibrosis compared to this study. They also reported minimal improvement among patients with severe hepatic fibrosis (grades II–III). Less than a quarter of patients had evidence of fibrosis regression after 24 months from baseline [26]. These Philippine studies on the impact of PZQ treatment on *S. japonicum*-related hepatic fibrosis, together with those carried out in China [13,30], repeatedly showed that PZQ had a limited antifibrotic effect on patients with severe fibrosis and that the US-based improvement in severity among patients with mild to moderate fibrosis after PZQ treatment might be secondary to its antiparasitic effect. PZQ kills the adult worms and the eggs, lowering the number of live ova deposited in the hepatic tissue that releases antigens. This decreases granuloma formation and eventually halts the hepatic fibrogenesis [7,10,26].

### 4.3. Serum miRNAs as Biomarkers of Hepatic Fibrosis

miR-150-5p, a dominant miRNA in the hematopoietic system [31], suppresses stress-induced tissue remodeling and fibrosis in the kidneys, lungs, and heart [32]. Deng et al. (2016) demonstrated the antifibrotic effect of miR-150 in cardiac fibroblasts of a mouse model of transverse aortic constriction by regulating c-Myb, a transcriptional activator of type I collagen gene [33]. In the lung tissue, overexpression of miR-150 suppressed hypoxia-induced formation and deposition of collagen fiber via the AKT/mTOR pathway [34]. The present study showed that circulating serum levels of miR-150 can detect the presence of US-detectable hepatic fibrosis among schistosomiasis patients. This is consistent with recent studies wherein circulating miR-150 levels were significantly lower in patients with schistosomiasis japonica-induced hepatic fibrosis compared to those with normal US findings [16,17].

let-7a-5p, a member of the let-7 miRNA family, has been reported to be downregulated in hepatic and hematologic samples from patients with non-tumorous fibrotic liver compared to healthy controls. Matsuura et al. showed that circulating let-7a levels in serum, plasma, and extracellular vesicles were positively correlated with the degree of hepatic fibrosis among patients with chronic hepatitis C [35,36]. It has been demonstrated that let-7a suppresses liver fibrosis in a mouse model via the TGFß/SMAD signal transduction pathway [35,36]. Overexpression of let-7a leads to lower cell viability and induced apoptosis in HSCs [37]. This study had similar findings to previous research where circulating serum levels of let-7a were significantly lower in schistosomiasis japonica patients with hepatic fibrosis as compared to patients with normal US findings [16]. This miRNA might represent a biomarker for stage-specific liver fibrosis regardless of etiology.

miR-146a-5p is a multi-functional miRNA shown to regulate multiple signal transduction pathways involved in fibrosis of various organs. It has been reported to have an antifibrotic effect in different mice models of liver fibrosis [38]. Zou et al. (2017a) showed that in vitro (human hepatocyte cell line L02) and in vivo (CCl_4_-treated rat hepatic fibrosis model) overexpression of miR-146a attenuate liver fibrosis and TGF-ß1-mediated epithelial-to-mesenchymal transition (EMT) of hepatocytes via targeting Smad4 [39]. Hepatocytes and HSCs undergo type 2 EMT into myofibroblasts, the primary cellular source of ECM deposited in scar tissues. Myofibroblasts can migrate to sites of injury where they deposit ECM that mainly consist of collagen [40]. The significantly lower level of serum miR-146a among schistosomiasis patients with hepatic fibrosis compared to those without hepatic fibrosis enrolled in this study was also observed in previous studies [16,17]. Further studies in Filipino patients are warranted to understand the differences in circulating miRNA levels in various liver diseases.

miR-93-5p is an antifibrotic factor shown to potentially regulate renal, cardiac, and hepatic fibrosis through the inhibition of TGF-ß1 pathway-associated epithelial–mesenchymal transition (EMT) and downregulation of genes involved in fibrogenesis [41]. Overexpression of miR-93 using miRNA mimics suppressed TGF-β1-induced EMT in HK2 cells (human kidney) [42] and ECM deposition in primary rat cardiac fibroblasts through the action of c-Ski, an inhibitory regulator of TGF-β signaling [43]. In this study, circulating serum miR-93 was significantly elevated in patients with US-detectable hepatic fibrosis as compared to those with normal US findings.

Aside from stage-specific diagnosis of schistosomiasis-induced hepatic fibrosis, it is important to identify biomarkers that can be used as prognostic factor for hepatic fibrosis risk. There are no previous reports on the use of miRNA levels in predicting outcomes of schistosomiasis-induced hepatic fibrosis after PZQ treatment in human [40]. Of the six selected miRNAs included in the current study, only the circulating serum level of miR-146 at baseline was significantly different between schistosomiasis patients with better outcome, namely those who remained without hepatic fibrosis, those who had regression of hepatic fibrosis, and those with poorer outcome of progressive fibrosis at 24 months from PZQ treatment. Patients with better outcomes had significantly higher baseline levels of miR-146 compared to those with poorer outcomes. Using a mouse model of schistosomiasis japonica, Cai et al. (2018) showed that levels of circulating serum miR-146a were decreased 6 weeks post-infection and then significantly increased 6 months after PZQ treatment [16]. Further studies are necessary to elucidate the potential role of miR-146 in hepatic fibrosis outcomes.

This study has several limitations: (1) the observation period was limited to 24 months after the baseline examination; (2) some changes in the US-detectable hepatic fibrosis among patients can be better appreciated in a longer and more frequent follow-up scheme [13]; (3) measurement was limited to six (6) miRNAs, which were selected based on literature review and on the preliminary PCR array study; and (4) only the strength of association but not the temporal relationship between hepatic fibrosis and differentially expressed circulating serum miRNAs was established using a baseline miRNA measurement.

## 5. Conclusions

In conclusion, this study showed that 42.6% of schistosomiasis patients had no fibrosis, 22.8% had mild fibrosis, and 34.6% had severe fibrosis at baseline. US-detectable hepatic fibrosis was prevalent in the study cohort during the time of examination. After 24 months, fibrosis reversal was observed in most patients with mild fibrosis but none in those with severe fibrosis at baseline. Patients with hepatic fibrosis had significantly decreased serum levels of the antifibrotic miR-146a, miR-150, and let-7a while having elevated levels of the profibrotic miR-93. The most likely candidate to distinguish between patients with and without hepatic fibrosis is miR-146a-5p. More research is required to confirm the effectiveness of this serum biomarker in predicting hepatic fibrosis in patients with schistosomiasis in the Philippines. Moreover, given the high prevalence of US-detectable hepatosplenic abnormalities in the study population, the formal inclusion of a community-based US evaluation in the human morbidity assessment component of the National Schistosomiasis Control and Elimination Program in the Philippines is recommended.

## Figures and Tables

**Figure 1 diagnostics-12-01902-f001:**
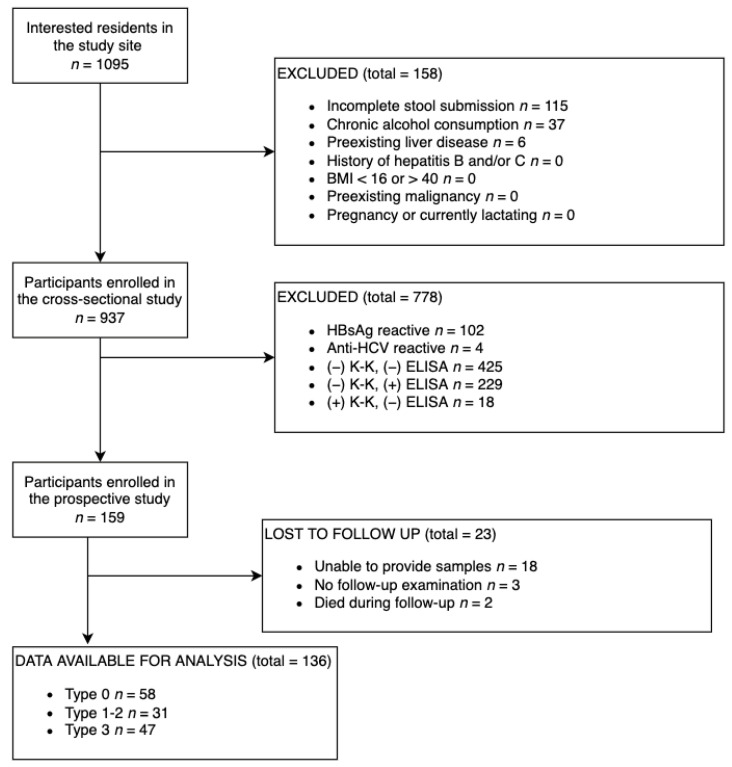
Participant recruitment and follow-up flow diagram.

**Figure 2 diagnostics-12-01902-f002:**
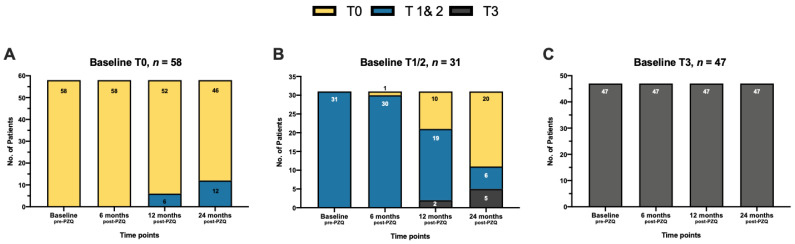
Dynamics of US-detectable hepatic fibrosis severity at different timepoints after PZQ treatment among the 136 patients enrolled in the prospective study. (**A**) Patients with no hepatic fibrosis at baseline (T0, *n* = 58). (**B**) Patients with mild hepatic fibrosis at baseline (T1/2, *n* = 31). (**C**) Patients with severe or typical network hepatic fibrosis at baseline (T3, *n* = 47). **Note**: Values presented are the absolute count of patients in each classification. **Abbreviation**: Praziquantel (PZQ).

**Figure 3 diagnostics-12-01902-f003:**
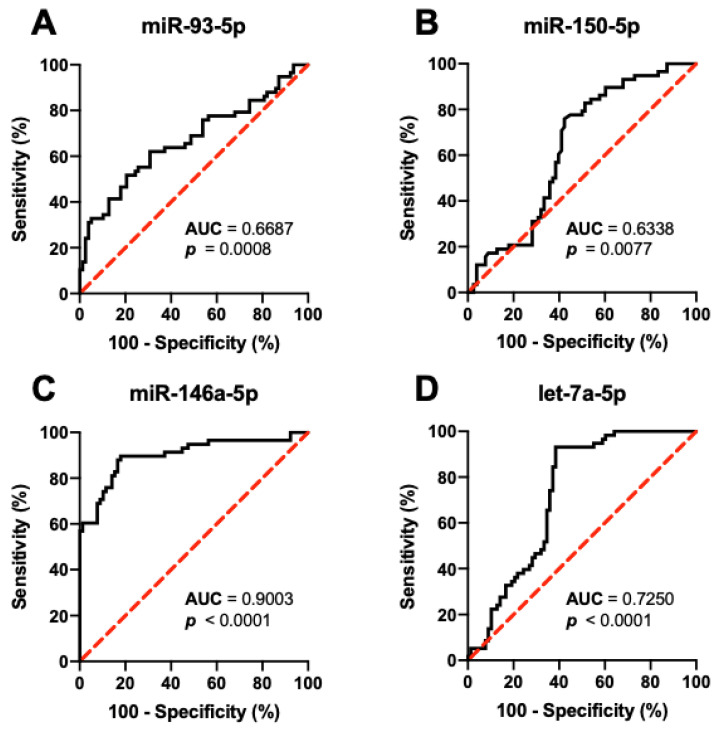
Differentiation of schistosomiasis-infected patients without ultrasonographic evidence of hepatic fibrosis (type 0) from those with hepatic fibrosis (types 1–3) by expression levels of circulating serum (**A**) miR-93-50, (**B**) miR-150-5p, (**C**) miR-146a-5-, and (**D**) let-7a-5p at baseline using the ROC curve analysis. **Note**: *p*-values were computed using Wilson/Brown test.

**Figure 4 diagnostics-12-01902-f004:**
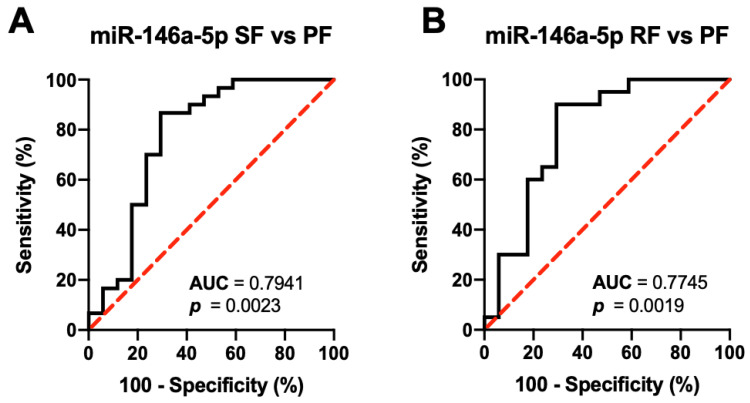
Differentiation of patients with progressive fibrosis (PF, *n* = 17) from those with (**A**) stable US findings (SF, *n* = 30) and (**B**) reversal of fibrosis (RF, *n* = 20) by baseline serum levels of the 6 target miRNAs. **Note**: *p*-values were computed using Wilson/Brown test.

**Table 1 diagnostics-12-01902-t001:** Baseline characteristics of the 136 patients enrolled in the prospective study.

Parameters	Schistosomiasis Patients ^1^			*p*-Value
	Type 0 *n* = 58	Types 1–2 *n* = 31	Type 3 *n* = 47	
Female	19	10	20	0.51 ^a, NS^
Male	39	21	27	
Age, mean ± SD (year)	36.9 ± 6.9	36.7 ± 5.6	37.8 ± 6.4	0.68 ^b, NS^
BMI ± SD (kg/m^2^)	28.7 ± 6.2	29.9 ± 5.9	26.1 ± 9.6	0.06 ^b, NS^
WBC, mean ± SD (4–10 × 10^9^/L)	8.7 ± 4.3	7.5 ± 3.7	9.7 ± 4.1	0.06 ^b, NS^
Hgb, mean ± SD (120–160 g/L)	151.0 ± 32.3	149.3 ± 19.7	143.9 ± 21.1	0.32 ^b, NS^
PLT, mean ± SD (150–450 × 10^9^/L)	327.3 ± 137.9	361.3 ± 157.2	384.8 ± 166.6	0.17 ^b, NS^
ALT, mean ± SD NV M: 21–72/F: 9–52 IU/L	35 ± 17.2	29 ± 11.2	32 ± 8.7	0.10 ^a, NS^
ALT abnormality no. (%) > 2 × ULN	6 (10.3)	4 (12.9)	6 (12.8)	0.9 ^a, NS^
AST, mean ± SD NV M: 17–59/F: 14–36 IU/L	31 ± 10.7	27 ± 6.3	29 ± 8.9	0.17 ^b, NS^
AST abnormality no. (%) > 2 × ULN	4 (6.9)	2 (6.5)	3 (6.5)	0.99 ^a, NS^
EPG, mean ± SD	89.2 ± 65.1	99.1 ± 52.3	157 ± 101.2	<0.01 ^b, c, S^
Infection Intensity ^2^				
Low	41	24	33	0.53 ^d, NS^
Moderate	16	5	10	
High	1	2	4	

**Note:** ^1^ Classification of hepatic fibrosis based on Ohmae et al. (1992), ^2^ infection intensity based on WHO, ^a^ Chi-square test, ^b^ Brown–Forsythe ANOVA test, ^c^ Dunnett’s T3 multiple comparisons test showed that EPG in Type 3 was significantly different to Type 0 (*p*-value < 0.0001) and Types 1–2 (*p*-value < 0.0001),^d^ Friedman test one-way ANOVA, ^S^ statistically significant at *p* < 0.05, ^NS^ not statistically significant. **Abbreviations**: Body mass index (BMI), white blood cell (WBC), hemoglobin (Hgb), platelet (PLT), alanine aminotransferase (ALT), aspartate aminotransferase (AST), egg per gram (EPG), Standard Deviation (SD).

**Table 2 diagnostics-12-01902-t002:** Descriptive characteristics of the 136 patients enrolled in the prospective study at baseline and 6, 12, and 24 months after PZQ treatment.

Parameters	Baseline No. (%)	Follow-Up					
		6 Months No. (%)	*p*-Value ^a^	12 Months No. (%)	*p*-Value ^a^	24 Months No. (%)	*p*-Value ^a^
***Schistosoma japonicum* positivity**							
3 stool K-K	136 (100)	0 (0.0)	-	0 (0)	-	34 (25.0)	-
EPG, mean ± SD	106 ± 45	-	-	-	-	46 ± 16	-
SEA ELISA	136 (100)	136 (100)	-	101 (74.3)	-	89 (65.4)	-
**Infection intensity ^1^**							
Low	98 (72.1)	0 (0)	-	0 (0)	-	32 (0)	-
Moderate	31 (22.8)	0 (0)		0 (0)		2 (0)	
High	7 (5.1)	0 (0)		0 (0)		0 (0)	
**Hepatic fibrosis ^2^**							
Type 0	58 (42.6)	59 (43.4)	-	63 (46.3)	-	66 (48.5)	-
Types 1 and 2	31 (22.8)	30 (22.1)		24 (17.6)		18 (13.2)	
Type 3	47 (34.6)	47 (34.6)		49 (36.0)		52 (38.2)	
**Portal vein wall thickness ^3^**							
normal	118 (86.8)	117 (86.0)	0.99 ^NS^	121 (89.0)	0.71 ^NS^	128 (94.1)	0.06 ^NS^
thickened	18 (13.2)	19 (14.0)		15 (11.0)		8 (5.9)	
**Liver size ^3^**							
normal	122 (89.7)	127 (93.4)	0.38 ^NS^	132 (97.1)	0.03 ^S^	133 (97.8)	0.01 ^S^
enlarged	14 (10.3)	9 (6.6)		4 (2.9)		3 (2.2)	
**Spleen size ^3^**							
normal	116 (85.3)	130 (95.6)	<0.01 ^S^	133 (97.8)	<0.01 ^S^	133 (97.8)	<0.01 ^S^
enlarged	20 (14.7)	6 (4.4)		3 (2.2)		3 (2.2)	

**Note**:^1^ Infection intensity classification based on WHO criteria, ^2^ classification of hepatic fibrosis based on Ohmae et al. (1992), ^3^ measurements based on Chinese population normal values, ^a^ pairwise comparison between baseline and follow-up using Fisher’s exact test at cut-off *p*-value of < 0.05. ^S^ statistically significant at *p*-value < 0.05. ^NS^ not statistically significant; **Abbreviations**: Kato–Katz (K-K), Soluble Egg Antigen ELISA (SEA ELISA), egg per gram (EPG), Standard Deviation (SD).

**Table 3 diagnostics-12-01902-t003:** Diagnostic performance in differentiating chronic schistosomiasis patients with and without hepatic fibrosis of the designated expression cut-off value at baseline for each circulating serum miRNA based on ROC analysis.

Target miRNA	AUC (95% CI)	Expression Level Cut-Off	Sensitivity % (95% CI)	Specificity % (95% CI)	LR
miR-146a-5p	0.90 (0.84–0.96)	<−8.16	60.34 (47.49–71.91)	98.72 (93.09–99.93)	47.07
let-7a-5p	0.73 (0.64–0.81)	<−8.75	5.172 (1.41–14.14)	98.72 (93.09–99.93)	4.03
miR-150-5p	0.64 (0.54–0.78)	<−7.67	12.07 (5.97–22.88)	96.15 (89.29–98.95)	3.14
miR-93-5p	0.67 (0.57–0.76)	>−3.42	13.79 (7.16–24.93)	98.72 (93.09–99.93)	10.76

**Abbreviations**: Area under the curve (AUC), likelihood ratio (LR), confidence interval (CI).

## Data Availability

The data that support the findings of this study are available in the Appendix A of this article.

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
