# Peer review of "Circulating microRNAs as Biomarkers of Hepatic Fibrosis in Schistosomiasis Japonica Patients in the Philippines"

_diagnostics, 2022, doi:10.3390/diagnostics12081902_

Round 1

Reviewer 1 Report

Hepatic fibrosis is the primary cause of morbidity and mortality in schistosomiasis, which affects more than 230 million people worldwide. Identification of novel biomarkers for liver fibrosis is crucial for the prognosis and therapy strategy for patients with schistosomiasis. The study from Tabios et al investigated the correlations between the host miRNA levels and the hepatic fibrosis severity and identified four miRNAs that can distinguish patients without ultrasonographic evidence of hepatic fibrosis (type 0) from those with hepatic fibrosis (types 1-3), with the miR-146a-5p showed the most promising potential. The authors also found that the circulating serum level of miR-146-5p at baseline was significantly different between schistosomiasis patients with better outcome and those with poorer outcome of progressive fibrosis at 24 months from PZQ treatment. This study is novel, interesting, and well-written. However, the manuscript suffers several weaknesses as listed below.

Major,

1. The authors did not emphasize their major findings in the conclusion section.

2. The authors did not describe why the Average Ct values of the endogenous SNORD95 and spiked-in UniSp6 were used to normalize the data, especially that cel-miR-39-3p miRNA mimic was also spiked to each sample during RNA extraction, which seems also for the purpose of data normalization. For the first-strand cDNA synthesis, was the same concentration of total RNA reverse-transcribed?

3. No details of SEA IgG ELISA described in the Methods.

Minor,

In the author list, Lydia R. Leonardo was marked with *, but not listed as a corresponding author in Line 19. Please check.

Line 80, a reference needed after ‘BALB/c mice’.

Line 108, MDA already appeared in Line 51

Line 179, Caenorhabditis elegans, italicized, and the amount of cel-miR-39-3p miRNA mimic added

Line 187, 1 μL enzyme mix, 0.5 μL synthetic RNA spike in UniSp6, 4.5 μL nuclease-free water? What’s the concentration of spiked-in UniSp6?

Table 2, in the line of SEA ELISA, the third column, 136 (100)?

Line 333, miR-146a-5p

Line 394, References needed after having fibrosis

Line 409, full name for ALD and NAFLD

Line 416, treatment, References needed here?

Line 434, adult worms and the eggs?

Line 452, positively correlated?

Line 453, References after chronic hepatitis C?

Line 464, Full name for EMT

Line 474, References needed after ‘in fibrogenesis’.

Line 476, ‘through c-Ski’ is unclear

Line 504-8, It reads like a non-integrated sentence.

References

The authors need carefully check all the references (orders and accuracy).

Line 424, Ref 24 should be: Ohmae H, Tanaka M, Hayashi M, et al. Improvement of ultrasonographic and serologic changes in Schistosoma japonicum-infected patients after treatment with praziquantel. The American Journal of Tropical Medicine and Hygiene. 1992 Jan;46(1):99-104. DOI: 10.4269/ajtmh.1992.46.99. PMID: 1536391.

Line 431, wrong citations 38, 39, are they ref 26 and 27, respectively?

Line 448, Reference 32 and 33 are the same as 15 and 16, respectively.

Author Response

Dear Editor,

We are very grateful for the comments of all our reviewers, and thank you for this opportunity to address them. We have made substantial changes to the manuscript and hope these changes satisfactorily meet your criteria for publication.

For your convenience, we have listed each comment along with our specific response below:

Reviewer 1

Hepatic fibrosis is the primary cause of morbidity and mortality in schistosomiasis, which affects more than 230 million people worldwide. Identification of novel biomarkers for liver fibrosis is crucial for the prognosis and therapy strategy for patients with schistosomiasis. The study from Tabios et al investigated the correlations between the host miRNA levels and the hepatic fibrosis severity and identified four miRNAs that can distinguish patients without ultrasonographic evidence of hepatic fibrosis (type 0) from those with hepatic fibrosis (types 1-3), with the miR-146a-5p showed the most promising potential. The authors also found that the circulating serum level of miR-146-5p at baseline was significantly different between schistosomiasis patients with better outcome and those with poorer outcome of progressive fibrosis at 24 months from PZQ treatment. This study is novel, interesting, and well-written. However, the manuscript suffers several weaknesses as listed below.

  • Thank you very much for taking the time to review our manuscript. We appreciate all your valuable inputs to improve our manuscript. Please check the attached manuscript with tracked changes.

Major,

  1. The authors did not emphasize their major findings in the conclusion section.
  • We added the main findings of the study in conclusion.

  1. The authors did not describe why the Average Ct values of the endogenous SNORD95 and spiked-in UniSp6 were used to normalize the data, especially that cel-miR-39-3p miRNA mimic was also spiked to each sample during RNA extraction, which seems also for the purpose of data normalization. For the first-strand cDNA synthesis, was the same concentration of total RNA reverse-transcribed?
  • Ideally, we should have included all three in the data normalization as suggested by Reviewer 1. However, we were not able to procure during the time of the experiment the primers for cel-miR-39-3p due to limited funding.
  • No, we did use the same concentration of total RNA for the cDNA synthesis. According to the product insert of the kit, the volume of the RNA extract is used for input amount in the cDNA synthesis since RNA concentration in extractions from serum and plasma cannot be accurately determined. We used the same volume of RNA extract (2uL for the 10 uL cDNA synthesis reaction) as input amount for the cDNA synthesis.

  1. No details of SEA IgG ELISA described in the Methods.
  • We added the details about the SEA IgG ELISA in the Methods section.

Minor,

In the author list, Lydia R. Leonardo was marked with *, but not listed as a corresponding author in Line 19. Please check.

  • This was corrected in the manuscript.

Line 80, a reference needed after ‘BALB/c mice’.

  • The correct citation was added.

Line 108, MDA already appeared in Line 51

  • This was corrected in the manuscript.

Line 179, Caenorhabditis elegans, italicized, and the amount of cel-miR-39-3p miRNA mimic added

  • This was corrected in the manuscript.

Line 187, 1 μL enzyme mix, 0.5 μL synthetic RNA spike in UniSp6, 4.5 μL nuclease-free water? What’s the concentration of spiked-in UniSp6?

  • 075 fmol/μL or 5x10^7 copies/μL. This information was added in the methods.

Table 2, in the line of SEA ELISA, the third column, 136 (100)?

  • This was corrected in the manuscript.

Line 333, miR-146a-5p

  • This was corrected in the manuscript.

Line 394, References needed after having fibrosis

  • This is part of the results of this study.

Line 409, full name for ALD and NAFLD

  • The definition was indicated in the manuscript.

Line 416, treatment, References needed here?

  • The correct citation was added.

Line 434, adult worms and the eggs?

  • This was corrected in the manuscript.

Line 452, positively correlated?

  • The study showed that increased let-7a levels in serum, plasma, and extracellular vesicles correlated with higher degree of hepatic fibrosis among patients with chronic hepatitis C.

Line 453, References after chronic hepatitis C?

  • The correct citations were added.

Line 464, Full name for EMT

  • The meaning of EMT was added in the manuscript.

Line 474, References needed after ‘in fibrogenesis’.

  • The correct citation was added.

Line 476, ‘through c-Ski’ is unclear

  • This was clarified in the manuscript. We added that c-Ski is an inhibitory regulator of TGF-β signaling.

Line 504-8, It reads like a non-integrated sentence.

  • This was corrected in the manuscript.

References

The authors need carefully check all the references (orders and accuracy).

Line 424, Ref 24 should be: Ohmae H, Tanaka M, Hayashi M, et al. Improvement of ultrasonographic and serologic changes in Schistosoma japonicum-infected patients after treatment with praziquantel. The American Journal of Tropical Medicine and Hygiene. 1992 Jan;46(1):99-104. DOI: 10.4269/ajtmh.1992.46.99. PMID: 1536391.

  • This was already corrected in the manuscript.

Line 431, wrong citations 38, 39, are they ref 26 and 27, respectively?

  • This was already corrected in the manuscript.

Line 448, Reference 32 and 33 are the same as 15 and 16, respectively.

  • This was already corrected in the manuscript.

Thank you.

Reviewer 2 Report

This paper aims to describe the employment of circulating microRNAs in the serum of patients suffering from schistosomiasis, as biomarkers of the development of hepatic fibrosis.

Overall, this manuscript is well written, the work has been well designed, the data are original, the methodology employed is adequate, and the results are properly analyzed and described. Some limitations of the work are exposed in the text. In this work, the hepatic fibrosis has been evaluated employing an ultrasound-based technique. Even though this technique is not the gold standard employed in the clinic of other countries, the paper is focused in the Philippines for the development of clinical techniques to be employed in development countries. The results described in the paper would be relevant for scientist working in liver disease, more precisely in liver fibrosis, especially in countries where the Schistosomiasis Japonica are considered an endemic disease. Thus, the present work might be employed in order to develop diagnostic tools, and for the control and prevention of the disease in those development countries.

In my opinion, this work could be acceptable for Diagnostics. There are only minor points to address in the manuscript.

-       Representative images from lesioned liver (with different stages of the disease) employing the ultrasound technique would be appreciated.

-       In page 2 lines 84 and 85 “In recent years, the prevalence and infection intensity of active schistosomiasis in the Philippines has decreased due to regular mass drug administration”. This concept has been previously described in Page 2 lines 50 and 51. It results redundant.  

-       In page 3 line 108, the abbreviation “MDA” has been previously described in line 51

-       The list of the miRNA specific primers employed in the work would be appreciated (even as Supplementary material).

-       The commercial supplier of ROX (Line 204 of page 5) is missing.

-       The information of the footnote of Figure 1 is irrelevant. It is probably a mistake.

-       Page 10 line 357. The text mentions “Table 10”. There is not table 10 is the work.  

The Supplementary material of this work is not available for this referee, so it has not been evaluated.

Author Response

Reviewer 2

This paper aims to describe the employment of circulating microRNAs in the serum of patients suffering from schistosomiasis, as biomarkers of the development of hepatic fibrosis. Overall, this manuscript is well written, the work has been well designed, the data are original, the methodology employed is adequate, and the results are properly analyzed and described. Some limitations of the work are exposed in the text. In this work, the hepatic fibrosis has been evaluated employing an ultrasound-based technique. Even though this technique is not the gold standard employed in the clinic of other countries, the paper is focused in the Philippines for the development of clinical techniques to be employed in development countries. The results described in the paper would be relevant for scientist working in liver disease, more precisely in liver fibrosis, especially in countries where the Schistosomiasis Japonica are considered an endemic disease. Thus, the present work might be employed in order to develop diagnostic tools, and for the control and prevention of the disease in those development countries.

  • Thank you very much for taking the time to review our manuscript. We appreciate all your valuable inputs to improve our manuscript. Please check the attached manuscript with tracked changes.

In my opinion, this work could be acceptable for Diagnostics. There are only minor points to address in the manuscript.

-       Representative images from lesioned liver (with different stages of the disease) employing the ultrasound technique would be appreciated.

  • This was included in Supplementary Figure 2.

-       In page 2 lines 84 and 85 “In recent years, the prevalence and infection intensity of active schistosomiasis in the Philippines has decreased due to regular mass drug administration”. This concept has been previously described in Page 2 lines 50 and 51. It results redundant.  

  • We removed this redundant statement.

-       In page 3 line 108, the abbreviation “MDA” has been previously described in line 51

  • This was corrected in the manuscript.

-       The list of the miRNA specific primers employed in the work would be appreciated (even as Supplementary material).

  • This was provided in Supplementary Table 1.

-       The commercial supplier of ROX (Line 204 of page 5) is missing.

  • Invitrogen™ ROX Reference Dye. This information was added in the methods.

-       The information of the footnote of Figure 1 is irrelevant. It is probably a mistake.

  • Thank you for pointing this out. We corrected the figure legend for Figure 1.

-       Page 10 line 357. The text mentions “Table 10”. There is not table 10 is the work.  

  • This should be Table 3. This was already corrected in the manuscript.

The Supplementary material of this work is not available for this referee, so it has not been evaluated.

  • We apologize for this. We made sure that the supplementary files were uploaded to the system. We are reuploading all the files if the reviewer needs to review them.

Thank you.